# The prognostic value of peripheral total and differential leukocyte count in renal progression: A community-based study

Chiung-Hui Yen[1,2], I-Wen Wu[3,4,5], Chin-Chan Lee[3,4,5], Kuang-Hung Hsu[6,7,8,9], Chiao-Yin Sun[3,4,5], Chun-Yu Chen[3,4,5], Heng-Chih Pan[3,4,5], Heng Jung Hsu [3,4,5,10]*

1 Department of Pediatrics, Taipei Medical University Hospital, Taipei, Taiwan, 2 College of Medicine, Taipei Medical University, Taipei, Taiwan, 3 Division of Nephrology, Chang Gung Memorial Hospital, Keelung, Taiwan, 4 College of Medicine, Chang Gung University, Tao-Yuan, Taiwan, 5 Community Medicine Research Center, Keelung Chang Gung Memorial Hospital, Keelung, Taiwan, 6 Healthy Aging Research Center, Chang Gung University, Taoyuan, Taiwan, 7 Laboratory for Epidemiology, Department of Health Care Management, Chang Gung University, Taoyuan, Taiwan, 8 Department of Emergency Medicine, Chang Gung Memorial Hospital, Taoyuan, Taiwan, 9 Department of Urology, Chang Gung Memorial Hospital, Taoyuan, Taiwan, 10 The Graduate Institute of Clinical Medical Sciences, Taoyuan School of Medicine, Chang Gung University Medical College, Taoyuan, Taiwan

* hsuaaron@gmail.com

**Data Availability Statement:** All relevant data are within the paper and its Supporting Information files.

## Abstract

### Background

Systemic inflammation is related to chronic kidney disease (CKD) patients. Elevated peripheral leukocyte count may be a herald of increased systemic inflammation and subclinical disease. Inflammation plays an important role in renal progression. The pattern of total and differential leukocyte count in CKD is not well understood. Besides, the association between total and differential leukocyte count and renal progression is still uncertain.

### Methods

We conducted a community-based cohort study with a follow-up period of two years to evaluate the total and differential leukocyte counts and renal progression association.

### Results

In our study population from the community with a total number of 2128, we found 15.7% (335/2128) CKD patients with a mean estimated glomerular filtration rate (eGFR) around 96 ± 26 ml/min/1.73 $m^2$. The peripheral total leukocyte count and also differential leukocyte count were significantly negatively correlated with eGFR. A total of 56 patients (3%) experienced a rapid progression of the kidney with the definition of eGFR reduction changes of 30% or greater within two years. Univariate analysis indicated that rapid renal progression was significantly associated with male gender, co-morbidity of diabetes mellitus (DM), higher uric acid levels, higher peripheral neutrophil, monocyte, and eosinophil counts. However, only the peripheral neutrophil count was positively and independently associated with rapid renal progression after multivariate analysis. The ROC curve analysis found that the

**Funding:** This investigation was partially supported by a grant from Chang Gung Medical Foundation Chang Gung Memorial Hospital, Keelung (CMRPG260362-3, CMRPG2A0451-3, CMRPG2B0151-3, CLRPG2J0011, and CMRPG2E0251-3). The funders had no role in study design, data collection, and analysis, decision to publish, or preparation of the manuscript. All the funding of support was received during this study. There was no additional external funding received for this study.

**Competing interests:** The authors have declared that no competing interests exist.

optimal cutoff value of peripheral neutrophil count for rapid progression was 2760/ mm$^3$, with an area under the curve of 0.813.

## Conclusion

Hyperinflammation with higher peripheral total and differential leukocyte count was noted in CKD patients. The peripheral neutrophil count was the only independent factor significantly associated with rapid renal progression. The optimal cutoff point of the peripheral neutrophil count with 2760/mm$^3$ is useful for determining the high-risk population for rapid renal progression with a satisfying sensitivity and specificity.

## Introduction

Chronic kidney disease (CKD) patients had a relatively worse prognosis, with a higher mortality rate due to cardiovascular deaths, infection, and stroke [1]. Inflammation is related to CKD with the characteristic of persistent low-grade chronic systemic inflammation [2]. These typical connections of malnutrition, inflammation, and atherosclerosis in CKD patients have been named malnutrition-inflammation-atherosclerosis syndrome (MIA syndrome) [3]. CKD patients with dialysis had a higher prevalence of protein-energy malnutrition and inflammation, with pathophysiology related to nutrient loss, increased protein catabolism, and hypoalbuminemia. Proinflammatory cytokines like interleukin -1 and tumor necrosis factor-alpha play a major role in the onset of metabolic alterations in CKD patients. Whether inflammation is either a trigger or a result of CKD is still complicated. However, many elements contribute to the inflammatory status of CKD, including increased production of proinflammatory cytokines, oxidative stress and acidosis, chronic and recurrent infections, altered metabolism of adipose tissue, and gut microbiota dysbiosis [2].

A complete blood count is commonly evaluated in patients with or without CKD for a routine examination. The inflammation status can easily be recognized by using the leukocyte total and also differential count analysis. Total leukocyte count increases significantly in response to infection, trauma, inflammation, and certain diseases. Recent findings suggested that elevated leukocyte count within the normal range, especially neutrophil and monocyte counts, maybe a herald of increased systemic inflammation and subclinical disease, such as cardiovascular disease [4–10] and type 2 DM [11–15]. Furthermore, elevated leukocyte count was associated with increased all-cause mortality [16–19]. The pattern of total and differential leukocyte count in CKD is not well understood. Besides, inflammatory processes may be critical in the development of kidney disease. The predictive value of total and differential leukocyte count for renal progression is still unclear. Rajiv et al. found that CKD patients had more eosinophil and granulocyte count and fewer lymphocytes counts than non-CKD patients, but with prominent variation [20]. They found that spikes in granulocyte and monocyte percentages in CKD are associated with end-stage renal disease (ESRD) and mortality [20]. However, Anam et al. found that peripheral eosinophilia is an independent predictor of tissue eosinophilia and subsequent progression to ESRD [21].

Due to the pattern of the total and differential leukocyte count in CKD and predictive value for renal progression are still unclear, we conduct this community-based cohort study with a follow-up period of two years to evaluate the association between total and differential leukocyte count and renal progression.

## Materials and methods

### Patient setting, data description, and study design

This study originated from a community-based survey for peripheral leukocyte counts and CKD. It was performed in the northeastern region of Taiwan, including the Wanli, Ruifang, Gongliao, and Anle districts, from March 2014 to November 2018. The inclusion criteria were age older than 30 years and the absence of pregnancy. The exclusion criteria were conditions that would interfere with peripheral leukocyte counts, such as current infection-related admission, chronic infection due to tuberculosis, arthritis, inflammatory bowel diseases, connective tissue disorders, other inflammatory diseases, or malignancy. Participants were also excluded if they were currently or had recently (within one month) received medicines that would interfere with peripheral leukocyte counts, such as steroid or immunosuppressant treatment. This community-based cohort study was designed to determine the association of baseline peripheral leukocyte counts with renal progression. The study population was followed up for two consecutive years in the community to evaluate the renal function change. The demographic survey assessed the medical history of systemic diseases, such as DM, hypertension, coronary artery disease (CAD), asthma, and cerebral vascular accident (CVA). Besides, the primary causes of CKD were also obtained.

Blood samples were obtained after an overnight fast. The following parameters were determined: complete blood cell count, renal biochemistry parameters, lipid profile, serum levels of uric acid, calcium, phosphate levels, and high sensitivity C-reactive protein (hs-CRP). Besides, the daily proteinuria amount was estimated by spot urine protein-to-creatinine ratio (mg/g) according to the NKF/DOQI [22]. The study was conformed to the Declaration of Helsinki's ethical guidelines and was performed with the Keelung Chang Gung Memorial Hospital's ethical committee's approval. The Institutional Review Board of the Chang-Gung Memorial Hospital approved this research (IRB No: 201102245A3). All participants agreed to join the study and signed an informed consent form before enrollment into the study.

### Definitions

The National Kidney Foundation defined CKD: K/DOQI classification for CKD and was determined to have persistent proteinuria or a decreased eGFR of less than 60 mL/min/1.73 m2, determined by the abbreviated Modification of Diet in Renal Disease equation [23]. Proteinuria was determined if urine albumin-to-creatinine ratio > 30 g/g or urine protein-to-creatinine ratio > 150 g/g. DM was defined as a fasting glucose level $\geq$ of 126 mg/dL or any hypoglycemic medication. Hypertension was considered if the patient received medical therapy for such a condition or if blood pressure was >140/90 mmHg. Smoking indicated any sustained past or current behaviors. Body mass index (BMI) was calculated as the weight in kilograms divided by the square of the height in meters. Rapid renal progression was defined as eGFR reduction changes of 30% or greater within two years [24].

### Statistical methods

For continuous variables, the values are expressed as the means and the standard deviations. T-test was applied for comparing the mean values of the two samples. One-way ANOVA was used for comparing the mean values of multiple samples. Categorical data were analyzed with the chi-square test or the Fisher exact test as appropriate. All statistical tests were 2-tailed. A P-value of < 0.05 was considered to indicate a statistically significant difference. Pearson or Spearman correlation coefficients were appropriately used to test the correlation between peripheral total and differential leukocyte count with eGFR. Logistic regression analysis was applied to identify

the association between these variables with the outcome of interest after adjusting for potential confounders, such as age, gender, and co-variants selection based on univariate analysis with variables with p < 0.1. We used a stepwise regression model (forward selection) with adjustment of the above co-variants. Conditional logistic regression analysis was performed to evaluate the odds ratio of factors associated with outcome. A receiver operating characteristic (ROC) curve was used to find the optimal cutoff point, pre-determined with sensitivity and specificity > 0.6 in peripheral neutrophil counts, serum creatinine levels, and daily proteinuria amount. Data were analyzed using SPSS 17.0 for Windows (SPSS Inc., Chicago, IL).

## Results

### The demography and primary data in our study population

In our study population from the community, the mean age was around 57 ± 13 years old, and 31% in men. The mean BMI was about 24.1 ± 3.8 Kg/m$^2$. The smoker population was around 21%. The percentage of NSAID usage and the Chinese herb of our study population was about 9% and 11%, respectively. The prevalence of co-morbidity of DM and hypertension is around 14% and 24%, respectively.

Among our study population of total number 2128, the mean eGFR was around 96 ± 26 ml/min/1.73 m$^2$ and we found 15.7% CKD patients (N = 335) and the prevalence of CKD stage 1, stage 2, stage 3, and stage 4–5 was about 5.2%, 4.4%, 5.5%, and 0.6%, respectively. The nutrition status was relatively fine, with a mean serum albumin level of 4.7 ± 0.3 g/dL. The mean peripheral leukocyte counts were around 5.4 ± 2.4 K/mm$^3$, and the neutrophil counts were predominant with a value of about 2.8 ± 1.5 K/mm$^3$, followed by lymphocyte counts with a value of about 1.8 ± 1.2 K/mm$^3$. The monocyte and eosinophil counts were relatively small, with mean counts around 284 ± 217/mm$^3$ and 133 ± 128/mm$^3$.

### Demographic data of non-CKD and CKD patients with different CKD stages

The mean age of CKD patients was older than the non-CKD population, and there is a significant positive relationship between the mean age and CKD stages (p < 0.001) (Table 1). Regarding gender difference, the male population percentage was significantly higher in more advanced CKD stages than earlier CKD stages and non-CKD population (p < 0.001). The BMI was similar between different CKD stages, but the BMI in CKD is significantly higher than in the non-CKD population (p < 0.001). The prevalence of smokers was also similar between different CKD stages, but the prevalence of smokers was higher in CKD than in non-CKD populations (p < 0.001.) The prevalence of using NSAID was also significantly higher in more advanced CKD stages than early CKD stages, except CKD stage 4 and 5 (p = 0.002). The prevalence of using Chinese herbs was similar between non-CKD and CKD patients with different stages. About the co-morbidity, the prevalence of DM, hypertension, CAD, and CVA was significantly in line with the progression of the CKD stages (p < 0.001), but the prevalence of asthma was similar between non-CKD and CKD patients with different stages. About the CKD primary causes, diabetic nephropathy was the main cause in advanced CKD patients (Stage 3–5), whereas unknown causes were the principal causes in early CKD patients (Stage 1–2).

### Laboratory data of non-CKD and CKD patients with different CKD stages

The hemoglobin and serum levels of albumin were significantly lower in CKD patients than non-CKD patients, and the hemoglobin and serum albumin levels were significantly negatively correlated with CKD stages (p < 0.001) (Table 2). The uric acid levels were significantly higher

**Table 1. Demographic data of non-CKD and CKD patients with different CKD stages.**

| | Non-CKD | CKD stage 1 | CKD stage 2 | CKD stage 3 | CKD stage 4 and 5 | P value$^\$$ |
|---|---|---|---|---|---|---|
| | N = 1793 | N = 110 | N = 95 | N = 118 | N = 12 | |
| Age (years) | 57 ± 12[£] | 54 ± 12 | 67 ± 11 | 72 ± 9 | 73 ± 12 | < 0.001* |
| Male gender (%) | 509 (28%)[£] | 35 (32%) | 45 (47%) | 57 (48%) | 5 (42%) | < 0.001* |
| BMI (Kg/m$^2$) | 23.8 ± 3.7[£] | 25.4 ± 4.4 | 25.1 ± 4.4 | 25.7 ± 3.6 | 25.1 ± 4.0 | 0.215 |
| Smoking (%) | 350 (20%)[£] | 34 (31%) | 30 (32%) | 29 (25%) | 3 (27%) | 0.022 |
| NSAID use (%) | 145 (8%)[£] | 9 (8%) | 11 (13%) | 24 (21%) | 0 (0%) | 0.002* |
| Chines Herb use (%) | 124 (11%) | 12 (16%) | 6 (10%) | 7 (9%) | 1 (13%) | 0.699 |
| Co-morbidity | | | | | | |
| Diabetes mellitus (%) | 187 (10%)[£] | 35 (32%) | 32 (34%) | 44 (37%) | 7 (58%) | < 0.001* |
| Hypertension (%) | 347 (20%)[£] | 34 (31%) | 48 (51%) | 68 (59%) | 11 (92%) | < 0.001* |
| CAD (%) | 130 (7%)[£] | 5 (5%) | 13 (14%) | 28 (24%) | 2 (17%) | < 0.001* |
| Asthma (%) | 55 (3%) | 4 (4%) | 1 (1%) | 2 (2%) | 1 (2%) | 0.512 |
| CVA (%) | 13 (1%)[£] | 1 (1%) | 2 (2%) | 6 (5%) | 1 (8%) | < 0.001* |
| Primary Causes of CKD | | | | | | < 0.001* |
| Diabetes (%) | | 34 (31%) | 32 (34%) | 44 (37%) | 6 (50%) | |
| Hypertension (%) | | 21 (19%) | 21 (22%) | 40 (34%) | 4 (33%) | |
| Chronic GN (%) | | 0 (0%) | 8 (8%) | 4 (3%) | 2 (17%) | |
| Unknown (%) | | 55 (50%) | 34 (36%) | 30 (25%) | 0 (0%) | |

Notes: Data are presented as mean ± standard deviation and number (%).

[£]Comparison between non-CKD and CKD patients by student T-test or chi-square test with a p-value less than 0.05.

$^\$$Comparison among CKD patients with different CKD stages by One-Way ANOVA with a linear trend.

*p-value < 0.05.

Abbreviations: CKD, chronic kidney disease; BMI, body mass index; NSAID, nonsteroidal anti-inflammatory drug; CAD, coronary artery disease; CVA, cerebral vascular accident, GN, glomerulonephritis.

in CKD patients than non-CKD patients and positively correlated with CKD stages (p < 0.001). The serum calcium and cholesterol levels were similar between non-CKD and CKD with different stages. The serum levels of phosphate were significantly higher in CKD patients than non-CKD patients, especially in CKD stage 4 and 5. Proteinuria amount was significantly higher in CKD patients than non-CKD patients and positively correlated with CKD stages (p < 0.001). The serum levels of hs-CRP were significantly higher in CKD patients than non-CKD patients and positively correlated with CKD stages (p = 0.045).

About the peripheral total and differential leukocyte counts, the total leukocyte counts were higher in CKD patients than non-CKD population (p < 0.001). The differential leukocyte counts, including neutrophil, lymphocyte, and monocyte counts, were also significantly higher in CKD patients than non-CKD populations (p < 0.001). However, the eosinophil counts were not significantly higher in CKD patients than non-CKD population (p = 0.06). Higher total leukocyte count levels were noted in more advanced CKD than earlier CKD, though without significance (p = 0.05). The differential leukocyte counts, including neutrophil, lymphocyte, monocyte, and eosinophil counts, were not significantly different across CKD stages. The correlation analysis between eGFR and the peripheral total and differential leukocyte counts was also performed to clarify the exact association between renal function and peripheral leukocyte count. The peripheral total leukocyte counts were significantly negatively correlated with eGFR (r = -0.102, p < 0.001) (Fig 1A). The differential leukocyte counts, including neutrophil, lymphocyte, and monocyte counts, were also negatively correlated with eGFR with

**Table 2. Laboratory data of non-CKD and CKD patients with different CKD stages.**

| | Non-CKD | CKD stage 1 | CKD stage 2 | CKD stage 3 | CKD stage 4 and 5 | P value$^$ |
|---|---|---|---|---|---|---|
| | N = 1793 | N = 110 | N = 95 | N = 118 | N = 12 | |
| **Laboratory data** | | | | | | |
| Albumin (g/dL) | $4.7 \pm 0.3^£$ | $4.7 \pm 0.2$ | $4.7 \pm 0.3$ | $4.6 \pm 0.4$ | $4.4 \pm 0.3$ | $< 0.001^*$ |
| eGFR (ml/min/1.73 m$^2$) | $100 \pm 23^£$ | $116 \pm 28$ | $76 \pm 8$ | $51 \pm 7$ | $22 \pm 6$ | $< 0.001^*$ |
| Uric acid (mg/dL) | $5.2 \pm 1.3^£$ | $5.3 \pm 1.3$ | $6.0 \pm 1.5$ | $6.7 \pm 1.6$ | $8.4 \pm 1.8$ | $< 0.001^*$ |
| hs-CRP (mg/L) | $2.5 \pm 5.3^£$ | $3.7 \pm 9.0$ | $3.9 \pm 14.2$ | $5.8 \pm 22.7$ | $6.1 \pm 10.8$ | $0.045^*$ |
| Hemoglobin (g/dL) | $13.5 \pm 1.5^£$ | $14.0 \pm 1.7$ | $13.6 \pm 1.8$ | $13.0 \pm 1.6$ | $10.4 \pm 1.6$ | $< 0.001^*$ |
| Calcium (mg/dL) | $9.3 \pm 0.3$ | $9.4 \pm 0.4$ | $9.4 \pm 0.4$ | $9.3 \pm 0.4$ | $9.3 \pm 0.4$ | 0.601 |
| Phosphate (mg/dL) | $3.8 \pm 0.5^£$ | $3.8 \pm 0.5$ | $3.7 \pm 0.6$ | $3.7 \pm 0.6$ | $4.5 \pm 0.7$ | $< 0.001^*$ |
| Cholesterol (mg/dL) | $211 \pm 38$ | $206 \pm 45$ | $208 \pm 41$ | $200 \pm 43$ | $213 \pm 49$ | 0.937 |
| Proteinuria (mg/day) | $63 \pm 23^£$ | $188 \pm 169$ | $415 \pm 632$ | $251 \pm 497$ | $1302 \pm 1610$ | $< 0.001^*$ |
| **Peripheral total and differential leukocyte count** | | | | | | |
| | | | | | | |
| Total leukocyte count (K/mm$^3$) | $5.2 \pm 2.2^£$ | $6.4 \pm 3.5$ | $6.1 \pm 3.1$ | $6.4 \pm 3.9$ | $6.6 \pm 3.3$ | 0.05 |
| Neutrophil count (K/mm$^3$) | $2.7 \pm 1.3^£$ | $3.5 \pm 2.0$ | $3.4 \pm 2.0$ | $3.6 \pm 2.4$ | $3.3 \pm 2.4$ | 0.255 |
| Lymphocyte count (K/mm$^3$) | $1.8 \pm 0.9^£$ | $2.3 \pm 1.9$ | $2.1 \pm 1.5$ | $2.4 \pm 2.3$ | $2.3 \pm 1.0$ | 0.195 |
| Monocyte count (/mm$^3$) | $269 \pm 187^£$ | $328 \pm 316$ | $391 \pm 345$ | $412 \pm 358$ | $299 \pm 137$ | 0.387 |
| Eosinophil count (/mm$^3$) | $130 \pm 125$ | $137 \pm 106$ | $165 \pm 190$ | $147 \pm 150$ | $170 \pm 167$ | 0.373 |

Notes: Data are presented as mean ± standard deviation.

$^£$Comparison between non-CKD and CKD patients by student T-test with a p-value less than 0.05.

$^$Comparison among CKD patients with different CKD stages by One-Way ANOVA with a linear trend.

$^*$p-value $< 0.05$.

Abbreviations: CKD, chronic kidney disease; eGFR, estimated glomerular filtration rate; hs-CRP, high sensitivity C-relative protein.

significance (Fig 1B–1D). Only the peripheral eosinophil count was not significantly correlated with eGFR (Fig 1E).

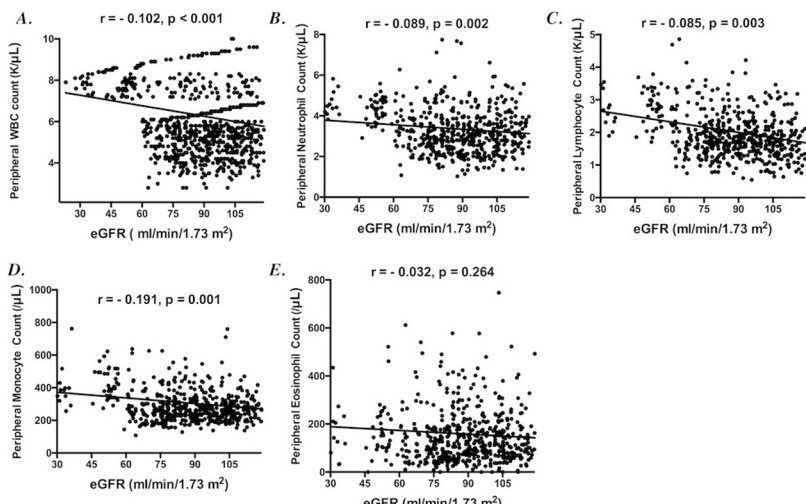

**Fig 1. The correlation between renal function (presented by estimated glomerular filtration rate) and peripheral total and differential leukocyte count in the study population.**

## Demographic and laboratory data of CKD patients with different CKD primary causes

In our study of CKD patients, diabetic nephropathy accounted for around 35% (116/335) of the primary causes of CKD. Hypertensive nephropathy comprised 26% (86/335) of the primary causes of CKD, and chronic glomerulonephritis (GN) covered 4% (14/335) of the primary causes of CKD. Unknown causes for CKD explained 36% (119/335) of the primary causes of CKD (Table 3). Those patients with chronic GN-related CKD were older than those with hypertensive nephropathy and diabetic nephropathy, and those CKD with unknown cause were significantly younger (P < 0.001). In addition, those CKD patients with unknown causes were significantly thinner than other primary causes of patients, and patients with diabetic nephropathy had significantly high BMI. The gender distribution was not significantly different between CKD patients with different primary causes. About laboratory data, lower levels of albumin and eGFR were noted in chronic GN than other causes of CKD. About inflammation markers, the levels of hs-CRP were similar between different primary causes of CKD. However, significantly higher peripheral leukocyte count, neutrophil count, and eosinophil count were noted in diabetic nephropathy than other primary causes of CKD. Besides, the percentage of rapid renal progression was similar between different primary causes of CKD (p = 0.569).

## Demographic and laboratory data of study population with and without rapid progression of kidney

Fifty-six patients (3%) experienced a rapid progression of kidney with a follow-up period of 2 years. Patients classified as rapid progression and non-rapid progression of kidney were

**Table 3. Demographic data and laboratory data of CKD patients with different primary causes.**

| Variables | Diabetic Nephropathy | Hypertensive Nephropathy | Chronic GN | Unknown Causes | P value$ |
|---|---|---|---|---|---|
|  | N = 116 | N = 86 | N = 14 | N = 119 |  |
| Age (years) | 66.2 ± 12.5 | 69.5 ± 11.3 | 70.5 ± 10.3 | 59.8 ± 15.4 | < 0.001* |
| Male gender (%) | 56 (48%) | 40 (47%) | 5 (36%) | 41 (35%) | 0.135 |
| BMI (Kg/m$^2$) | 26.9 ± 4.2 | 25.6 ± 4.2 | 25.1 ± 3.5 | 23.8 ± 3.4 | < 0.001* |
| Rapid progression (%) | 5 (4.3%) | 3 (3.5%) | 1 (7.1%) | 2 (1.7%) | 0.569 |
| **Laboratory data** |  |  |  |  |  |
| Albumin (g/dL) | 4.7 ± 0.4 | 4.6 ± 0.3 | 4.4 ± 0.4 | 4.7 ± 0.3 | 0.028* |
| eGFR (ml/min/1.73 m$^2$) | 75 ± 33 | 70 ± 27 | 58 ± 22 | 90 ± 36 | < 0.001* |
| Uric acid (mg/dL) | 6.4 ± 1.7 | 6.4 ± 1.7 | 6.7 ± 2.0 | 5.6 ± 1.4 | < 0.001* |
| hs-CRP (mg/L) | 6.6 ± 4.1 | 4.3 ± 13.3 | 5.4 ± 9.7 | 2.6 ± 6.2 | 0.353 |
| Total leukocyte count (K/mm$^3$) | 7.4 ± 3.3 | 6.5 ± 4.1 | 5.3 ± 2.1 | 5.3 ± 3.1 | < 0.001* |
| Neutrophil count (K/mm$^3$) | 4.1 ± 2.1 | 3.8 ± 2.4 | 2.4 ± 0.9 | 2.9 ± 1.8 | 0.004* |
| Lymphocyte count (K/mm$^3$) | 2.7 ± 1.5 | 2.4 ± 2.5 | 1.7 ± 0.6 | 1.9 ± 1.9 | 0.117 |
| Monocyte count (/mm$^3$) | 443 ± 331 | 398 ± 362 | 265 ± 104 | 298 ± 316 | 0.077 |
| Eosinophil count (/mm$^3$) | 193 ± 179 | 141 ± 131 | 90 ± 47 | 120 ± 127 | 0.031* |

Notes: Data are presented as mean ± standard deviation and number (%).

$Comparison among CKD patients with different primary causes by One-Way ANOVA.

*p-value < 0.05.

Abbreviations: CKD, chronic kidney disease; BMI, body mass index; eGFR, estimated glomerular filtration rate; hs-CRP, high sensitivity C-relative protein; GN, glomerulonephritis.

significantly different in numerous demographic and clinical variables (Tables 4 and 5). Compared with non-rapid kidney progression patients, those with rapid progression of kidney were more likely to be male, had DM as a co-morbidity, and had higher serum uric acid levels. In particular, higher peripheral total and differential leukocyte count levels, including neutrophil, lymphocyte, monocyte, and eosinophil count, were significantly higher in patients with rapid renal progression than non-rapid progression.

## Univariate and multivariate analysis of variables associated with rapid progression of kidney

The univariate analysis indicated that rapid renal progression was significantly associated with male gender, co-morbidity of DM, higher uric acid levels, and higher peripheral neutrophil, monocyte, and eosinophil counts levels. However, after multivariate analysis, rapid renal progression was only positively and independently significantly associated with peripheral neutrophil count. In contrast, male gender, co-morbidity of DM, and uric acid levels lost significance (Table 6).

## ROC analysis of the renal rapid progression in study population

As described above, the multivariate analyses revealed that higher levels of peripheral neutrophil count were associated with rapid renal progression. Consequently, ROC curve analyses, pre-determined with sensitivity and specificity > 0.6 to find the optimal point, were modified to estimate the predictive and cutoff values of rapid renal progression. The ROC curve analysis for the peripheral neutrophil counts, plasma creatinine levels, and proteinuria amounts is shown in Fig 2. The peripheral neutrophil counts had a higher area under the curve than other variables associated with rapid renal progression. The optimal cutoff value for the peripheral neutrophil count was 2760 /mm$^3$, with 91% sensitivity and 72% specificity for rapid renal progression associations. The area under the curve for the peripheral neutrophil count was 0.813

**Table 4. Demographic data of study population with and without rapid progression of kidney.**

| Variables | Without Rapid Progression | With Rapid progression | P-value |
|---|---|---|---|
| | N = 2067 | N = 56 | |
| Age (years) | 57.4 ± 12.5 | 57.5 ± 14.6 | 0.956 |
| Male gender (%) | 622 (30%) | 26 (46%) | 0.009* |
| BMI (Kg/m$^2$) | 24.1 ± 3.8 | 24.7 ± 3.9 | 0.244 |
| Smoking (%) | 426 (21%) | 17 (31%) | 0.069 |
| NSAID (%) | 186 (9%) | 3 (5%) | 0.353 |
| Chines Herb use (%) | 147 (11%) | 3 (9%) | 0.756 |
| Co-morbidity | | | |
| Diabetes mellitus (%) | 289 (14%) | 16 (29%) | 0.002* |
| Hypertension (%) | 487 (24%) | 19 (34%) | 0.08 |
| CAD (%) | 169 (8%) | 8 (14%) | 0.109 |
| Asthma (%) | 61 (3%) | 2 (4%) | 0.797 |
| CVA (%) | 21 (1%) | 2 (4%) | 0.071 |

Notes: Data are presented as mean ± standard deviation and number (%).

*p-value < 0.05.

Abbreviations: CKD, chronic kidney disease; BMI, body mass index; NSAID, nonsteroidal anti-inflammatory drug; CAD, coronary artery disease; CVA, cerebral vascular accident.

**Table 5. Laboratory data of study population with and without rapid progression of kidney.**

| Variables | Without Rapid Progression | With Rapid progression | P value |
|---|---|---|---|
| | N = 2067 | N = 56 | |
| Laboratory data | | | |
| Albumin (g/dL) | 4.7 ± 0.3 | 4.7 ± 0.3 | 0.854 |
| eGFR (ml/min/1.73 m$^2$) | 96 ± 26 | 93 ± 27 | 0.402 |
| Uric acid (mg/dL) | 5.3 ± 1.4 | 5.9 ± 1.7 | 0.003* |
| hs-CRP (mg/L) | 2.4 ± 8.4 | 3.3 ± 8.0 | 0.432 |
| Hemoglobin(g/dL) | 13.5 ± 1.6 | 13.7 ± 1.9 | 0.351 |
| Calcium, mg/dL | 9.3 ± 0.3 | 9.3 ± 0.4 | 0.755 |
| Phosphate, mg/dL | 3.8 ± 0.5 | 3.9 ± 0.5 | 0.102 |
| Cholesterol (mg/dL) | 210 ± 38 | 213 ± 50 | 0.454 |
| Proteinuria (mg/day) | 102 ± 248 | 137 ± 251 | 0.299 |
| Peripheral leukocyte count (K/mm$^3$) | 5.3 ± 2.4 | 7.3 ± 2.1 | < 0.001* |
| Peripheral Neutrophil count (K/mm$^3$) | 2.8 ± 1.4 | 4.5 ± 1.5 | < 0.001* |
| Peripheral Lymphocyte count (K/mm$^3$) | 1.8 ± 1.1 | 2.4± 1.2 | 0.014* |
| Peripheral Monocyte count (/mm$^3$) | 281 ± 214 | 421 ± 277 | 0.002* |
| Peripheral Eosinophil count (/mm$^3$) | 131 ± 127 | 191 ± 196 | 0.024* |

Notes: Data are presented as mean ± standard deviation.

*p-value < 0.05.

Abbreviations: CKD, chronic kidney disease; eGFR, estimated glomerular filtration rate; hs-CRP, high sensitivity C-reactive protein.

(95% confidence interval = 0.714–0.912). The optimal cutoff value for plasma creatine was difficult to find with sensitivity and specificity > 0.6, and an area under the curve was 0.566 (95% confidence interval = 0.431–0.70). The optimal cutoff value for proteinuria amount was also problematic to find with sensitivity and specificity > 0.6, and an area under the curve was 0.53 (95% confidence interval = 0.407–0.654).

**Table 6. Univariate and multivariate analysis of variables associated with rapid progression of kidney.**

| Variables | Univariate | | Multivariable | P-value |
|---|---|---|---|---|
| | OR (95% CI) | P-value | OR (95% CI) | |
| Age (year) | 1.001 (0.980–1.022) | 0.956 | 0.968 (0.937–1.000) | 0.051 |
| Male gender (yes vs. no) | 2.013 (1.181–3.433) | 0.01* | 0.791 (0.287–2.185) | 0.652 |
| Smoking (yes vs. no) | 1.705 (0.953–3.051) | 0.072 | | |
| Diabetes mellitus (yes vs. no) | 2.461 (1.36–4.453) | 0.003* | | |
| Hypertension (yes vs. no) | 1.646 (0.938–2.889) | 0.082 | | |
| CVA (yes vs. no) | 0.997 (0.993–1.000) | 0.09 | | |
| eGFR (ml/min/1.73 m$^2$) | 0.995 (0.985–1.006) | 0.401 | | |
| Uric acid (mg/dL) | 1.268 (1.081–1.487) | 0.004* | | |
| Neutrophil count (K/mm$^3$) | 1.446 (1.239–1.688) | < 0.001* | 1.357 (1.138–1.618) | 0.001* |
| Lymphocyte count (K/mm$^3$) | 1.238 (1.029–1.489) | 0.023* | | |
| Monocyte count (K/mm$^3$) | 4.503 (1.627–12.466) | 0.004* | | |
| Eosinophil count (K/mm$^3$) | 8.209 (1.228–54.871) | 0.03* | | |

Note: Multivariable analysis with backward selection model for all variables.

*p value < 0.05.

Abbreviations: OR: odds ratio; CI, confidence interval; BMI, body mass index; NSAID, nonsteroidal anti-inflammatory drug; eGFR, estimated glomerular filtration rate.

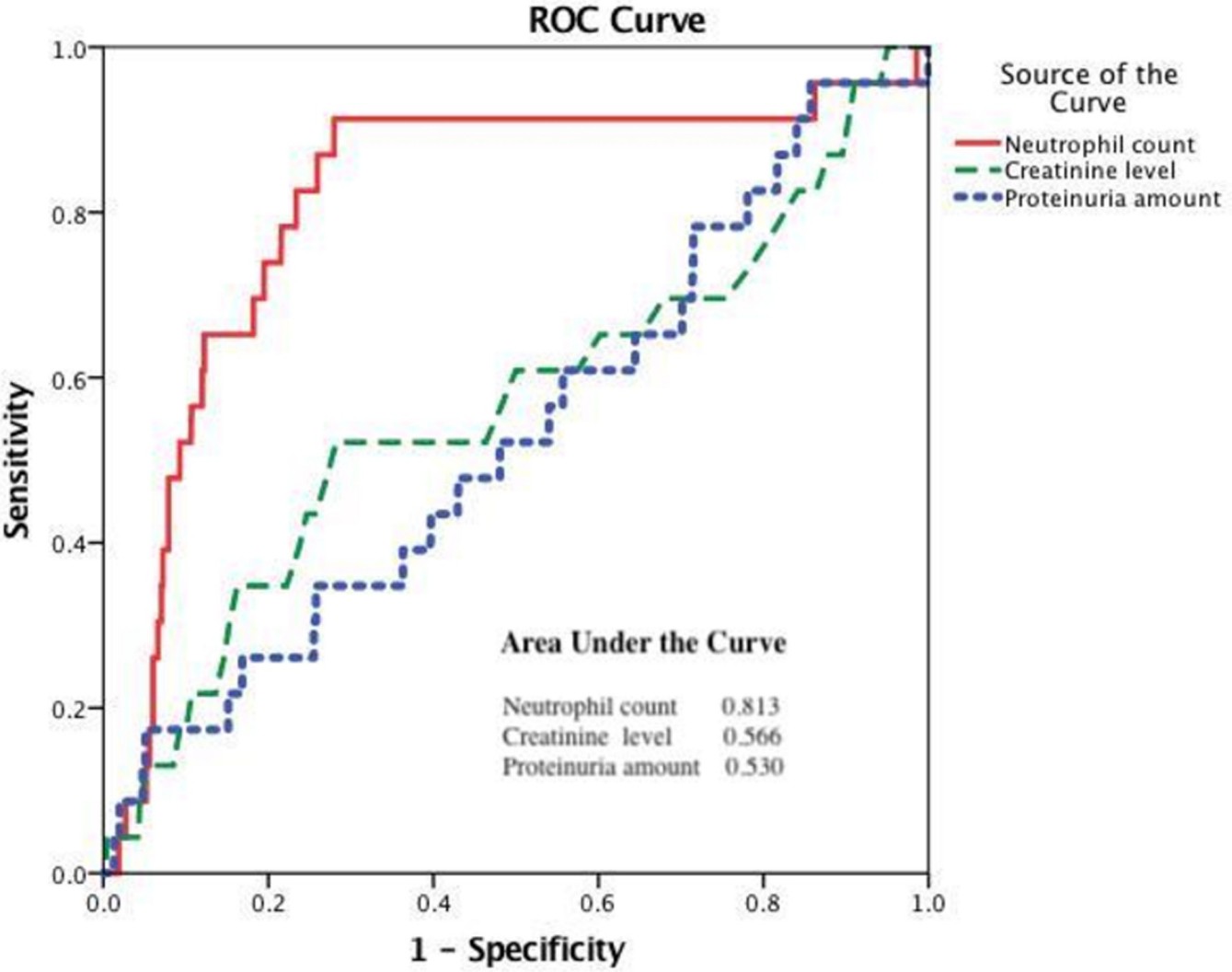

**Fig 2. The receiver operating characteristic curves and the area under the peripheral neutrophil count, serum creatinine, and proteinuria amount for discriminating study population with and without rapid progression.**

### Demographic and laboratory data of study population with lower and higher peripheral neutrophil counts

We divided the study population into lower and higher peripheral neutrophil count groups using the cutoff value of peripheral neutrophil count 2760 /mm$^3$. Higher peripheral neutrophil count group population were more likely to be male and smoker, had DM and hypertension as a co-morbidity, had higher BMI and serum levels of uric acid, hs-CRP, calcium, hemoglobin, and proteinuria, and had lower levels of eGFR and phosphate (Tables 7 and 8). In addition to peripheral neutrophil counts, peripheral total and differential leukocyte counts of lymphocyte, monocyte, and eosinophil counts were also significantly higher in the higher neutrophil count group than the lower group population. Moreover, those population with higher neutrophil group experienced a significantly higher eGFR reduction percentage than the lower neutrophil group (5.2 ± 17.7% vs. 1.2 ± 12.1%, p < 0.001) (Fig 3).

## Univariate and multivariate analysis of variables associated with higher neutrophil group

The univariate analysis indicated that the higher neutrophil group was significantly associated with male gender, higher BMI and smoker, co-morbidity of DM, hypertension, lower eGFR and phosphate levels, and higher uric acid, hs-CRP, calcium, hemoglobin, and proteinuria levels. However, the higher neutrophil group was only positively and independently significantly associated with the smoker, less age, higher BMI, co-morbidity of DM, and higher levels of uric acid, hs-CRP, and hemoglobin after multivariate analysis, whereas male gender, co-morbidity of hypertension, levels of eGFR, calcium, phosphate, and proteinuria lost significance (Table 9).

## Discussion

Our study found that peripheral total and differential leukocyte count levels, including neutrophil, lymphocyte, and monocyte, were significantly higher in CKD patients than in non-CKD populations. Besides, peripheral total leukocyte count and differential leukocyte count were significantly negatively correlated with eGFR. These results confirmed that hyperinflammation was prominent in CKD and was negatively correlated with renal function. Besides, those patients with rapid renal progression had higher total and differential leukocyte count than renal non-rapid progression patients. Among the peripheral total leukocyte count and differential leukocyte count, peripheral neutrophil count is independently associated with renal rapid-progression after multivariate logistic regression. This result showed that peripheral neutrophil count might play an important role in renal progression.

Our result is similar to a previous study by Agarwal et al., which found that the spikes in neutrophil percentages among CKD patients were independently associated with ESRD and death [20]. However, they did not find the association between an absolute neutrophil count or neutrophil percentage with CKD progression outcome. Instead, they used the neutrophil spike event with the definition of neutrophil percentage increase more than 70% during the follow-up period as a marker to be associated with renal progression. This difference may be due to the study design and also the study population with racial differences. Besides, study number differences may also account for the different results between these two studies. Our study population number (N = 2128) is larger than their study patients' number (N = 420). Besides, there are also several studies focusing on the neutrophil-to-lymphocyte ratio (NLR) and renal outcome. Kocyigit et al. suggested that NLR may predict the progression rate of CKD stage 4 to dialysis [25]. Besides, Yuan et al. also found that NLR is associated with ESRD risk in CKD stage 4 patients [26]. Nevertheless, Altunoren et al. found that NLR is an indicator of CKD inflammation and is not an independent predictor of CKD progression except in the more advanced CKD stage [27].

Our study found that peripheral neutrophil count was a significant predictor for rapid renal progression, defined as an eGFR reduction rate of more than 30% within two years. The peripheral neutrophil count of 2760/mm3 is a good cutoff point to determine the rapid renal progression with a sensitivity of 91% and specificity of 72%. This finding is useful to determine high-risk patients. The study found that peripheral neutrophil count was significantly an independent factor associated with rapid renal progression in the non-CKD and CKD population. Neutrophils may migrate to the kidney by increasing spontaneous adhesion, followed by abnormal activation of proinflammatory cytokines secretion, degranulated products, and reactive oxygen species, further damaging the kidneys. Clinical studies have found that total peripheral leukocytes, neutrophils, and the neutrophil-to-lymphocyte ratio (NLR) were independently associated with diabetic kidney disease in type 2 DM [28–33]. Our finding may

**Table 7. Demographic data of study population with lower and higher peripheral neutrophil counts.**

| | Lower Neutrophil group | Higher Neutrophil group | P-value |
|---|---|---|---|
| | N = 1235 | N = 2346 | |
| Age (years) | 57.9 ± 12.1 | 57.2 ± 13.5 | 0.116 |
| Male gender (%) | 302 (25%) | 883 (38%) | < 0.001* |
| BMI (Kg/m$^2$) | 24 ± 3.6 | 25 ± 3.8 | < 0.001* |
| Smoking (%) | 187 (15%) | 624 (27%) | < 0.001* |
| NSAID (%) | 113 (9%) | 210 (9%) | 0.857 |
| Chines Herb use (%) | 88 (11%) | 130 (9%) | 0.188 |
| Co-morbidity | | | |
| Diabetes mellitus (%) | 120 (10%) | 408 (17%) | < 0.001* |
| Hypertension (%) | 258 (21%) | 628 (27%) | < 0.001* |
| CAD (%) | 96 (8%) | 219 (9%) | 0.112 |
| Asthma (%) | 35 (3%) | 67 (3%) | 0.962 |
| CVA (%) | 14 (1%) | 30 (1%) | 0.703 |

Notes: Data are presented as mean ± standard deviation and number (%).

*p-value < 0.05.

Higher Vs. Lower Neutrophil group was divided by cutoff point with peripheral Neutrophil count 2.76 K/mm$^3$.

Abbreviations: CKD, chronic kidney disease; BMI, body mass index; NSAID, nonsteroidal anti-inflammatory drug; CAD, coronary artery disease; CVA, cerebral vascular accident.

raise questions about the clinical outcome between diabetes kidney disease (DKD) development and CKD development and progression. Our study found that DM is significantly associated with higher peripheral neutrophil levels in multivariate analysis (OR: 1.53, p = 0.001). It

**Table 8. Laboratory data of study population with lower and higher peripheral neutrophil counts.**

| Variables | Lower Neutrophil group | Higher Neutrophil group | P-value |
|---|---|---|---|
| | N = 1235 | N = 2346 | |
| Laboratory data | | | |
| Albumin (g/dL) | 4.7 ± 0.3 | 4.7 ± 0.3 | 0.141 |
| eGFR (ml/min/1.73 m$^2$) | 98 ± 25 | 95 ± 26 | 0.001* |
| Uric acid (mg/dL) | 5.1 ± 1.3 | 5.7 ± 1.4 | < 0.001* |
| hs-CRP(mg/L) | 1.5 ± 3.1 | 2.5 ± 7.2 | < 0.001* |
| Hemoglobin(g/dL) | 13.3 ± 1.5 | 13.8 ± 1.5 | < 0.001* |
| Calcium, mg/dL | 9.3 ± 0.3 | 9.4 ± 0.3 | < 0.001* |
| Phosphate, mg/dL | 3.83 ± 0.54 | 3.79 ± 0.53 | 0.037* |
| Cholesterol (mg/dL) | 209 ± 37 | 210 ± 40 | 0.749 |
| Proteinuria (mg/day) | 94 ± 244 | 122 ± 368 | 0.021* |
| Peripheral leukocyte count (K/mm$^3$) | 4.3 ± 0.8 | 6.4 ± 1.8 | < 0.001* |
| Peripheral Neutrophil count (K/mm$^3$) | 2.2 ± 0.4 | 3.8 ± 1.0 | < 0.001* |
| Peripheral Lymphocyte count (K/mm$^3$) | 1.7 ± 0.6 | 2.1 ± 0.9 | < 0.001* |
| Peripheral Monocyte count (/mm$^3$) | 238 ± 73 | 331 ± 165 | < 0.001* |
| Peripheral Eosinophil count (/mm$^3$) | 139 ± 144 | 165 ± 151 | < 0.001* |

Notes: Data are presented as mean ± standard deviation.

*p value < 0.05.

Abbreviations: CKD, chronic kidney disease; eGFR, estimated glomerular filtration rate; hs-CRP, high sensitivity C-reactive protein.

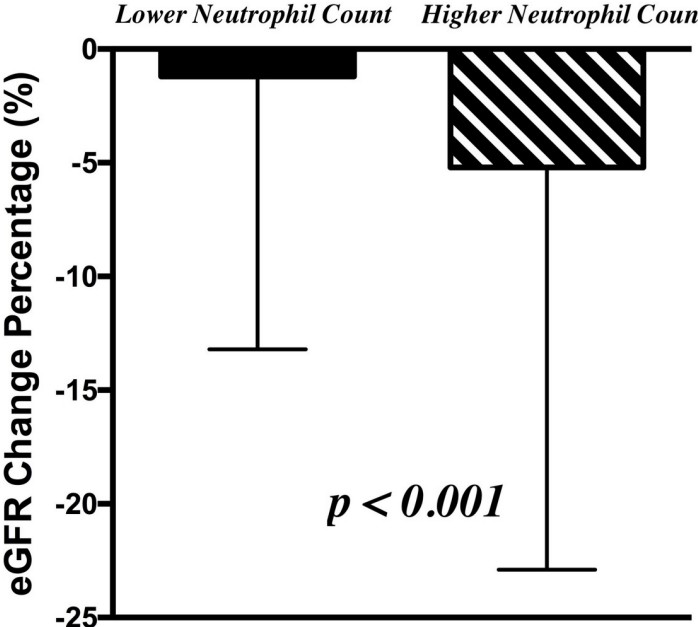

**Fig 3. The renal function (presented by estimated glomerular filtration rate) change during the 2-year follow-up period between the study population with the lower and higher neutrophil count.**

showed that higher neutrophil levels were prominently associated with DM. However, the multivariate logistic regression analysis showed that the peripheral neutrophil count was significantly associated with rapid renal progression after adjusting DM and other possible confounding factors. This finding suggests that neutrophil counts were a strong predictor of CKD progression than DKD development.

**Table 9. Univariate and multivariate analysis of variables associated with higher neutrophil group.**

| Variables | Univariate | P-value | Multivariable | P-value |
|---|---|---|---|---|
| | OR (95% CI) | | OR (95% CI) | |
| Age (year) | 0.996 (0.99–1.001) | 0.116 | 0.984 (0.978–0.991) | < 0.001* |
| Male gender (yes vs. no) | 1.865 (1.598–2.176) | < 0.001* | | |
| BMI (Kg/m$^2$) | 1.11 (1.088–1.132) | < 0.001* | 1.046 (1.021–1.071) | < 0.001* |
| Smoking (yes vs. no) | 2.038 (1.701–2.442) | < 0.001* | 1.492 (1.205–1.847) | < 0.001* |
| Diabetes mellitus (yes vs. no) | 1.958 (1.577–2.431) | < 0.001* | 1.53 (1.196–1.956) | 0.001* |
| Hypertension (yes vs. no) | 1.389 (1.178–1.639) | < 0.001* | | |
| eGFR (ml/min/1.73 m$^2$) | 0.995 (0.993–0.998) | 0.001* | | |
| Uric acid (mg/dL) | 1.385 (1.311–1.463) | < 0.001* | 1.236 (1.155–1.322) | < 0.001* |
| hs-CRP(mg/L) | 1.09 (1.054–1.128) | < 0.001* | 1.063 (1.03–1.097) | < 0.001* |
| Hemoglobin(g/dL) | 1.275 (1.217–1.335) | < 0.001* | 1.154 (1.091–1.222) | < 0.001* |
| Calcium(mg/dL) | 1.503 (1.218–1.856) | < 0.001* | | |
| Phosphate(mg/dL) | 0.871 (0.765–0.992) | 0.037* | | |
| Proteinuria(g/day) | 1.469 (1.039–2.077) | 0.03* | | |

Note: Multivariate analysis with backward selection model for all variables.

*p value < 0.05.

Abbreviations: CKD, chronic kidney disease; OR: odds ratio; CI, confidence interval; BMI, body mass index; eGFR, estimated glomerular filtration rate; hs-CRP, high sensitivity C-reactive protein.

Bowe et al. also evaluated the association between peripheral leukocyte count and risk of incident CKD and progression to ESRD [34]. This study enrolled more than 1594700 United States veterans with a median follow-up duration of 9.2 years and found that those with higher monocyte count had significantly higher risks of incident CKD and CKD progression to ESRD. Although our study did not enroll as many populations as Bowe et al. and the follow-up duration of two years was relatively not long enough, our study also found the peripheral monocyte counts were associated with rapid progression of the kidney. However, the association loss significance after multivariate analysis and only peripheral neutrophil counts was independently significantly associated with rapid progression of kidney. Due to Bowe et al. did not evaluate the role of peripheral neutrophils in the association of renal progression, the result of positive connection between monocyte counts and incident CKD and progression to ESRD is reasonable. However, the result of our study and Bowe et al. all suggested that inflammatory cell infiltrates may play a role in the development and progression of kidney disease.

Some reports showed that a higher eosinophil count was positively associated with advanced CKD stages [35]. Furthermore, Tariq et al. found that peripheral eosinophilia was an independent predictor of tissue eosinophilia and subsequent progression to ESRD [21]. Our study found that peripheral total and differential leukocyte counts, including neutrophil, lymphocyte, monocyte, and eosinophil counts, were significantly higher in patients with rapid renal progression than non-rapid progression. However, after multivariate binary logistic regression, only peripheral neutrophil counts were independently associated with rapid renal progression. Besides, our study found that peripheral eosinophil count was not significantly higher in CKD patients, and the absolute eosinophil count was not significantly correlated with eGFR. The similar but slightly different result of Tariq et al. from ours may be due to racial differences and study design. In the study of Tariq et al., they enrolled those patients who were receiving kidney biopsy to evaluate the association of peripheral eosinophilia (> 4% of peripheral blood leukocytes) with the risk of progression to ESRD. Those patients who need to receive kidney biopsy may be different from our study population from the community. Besides, the renal outcome endpoint is also different between the two studies. The most important point is that they did not adjust other differential leukocyte counts, which may discriminate the conclusion from our finding.

Our study evaluated the factors associated with peripheral neutrophil counts and found many factors such as male gender, higher BMI, smoking, presence of DM and hypertension, lower eGFR and phosphate levels, higher levels of uric acid, hs-CRP, calcium, hemoglobin, and proteinuria were associated with a higher neutrophil count. Among the above factors associated with the higher neutrophil count, we found that lots of proven poor prognostic factors for renal progression were significantly associated with the higher neutrophil count, such as higher BMI, smoking, presence of DM and hypertension, lower eGFR levels, higher levels of uric acid and proteinuria. It may explain why peripheral neutrophil count was the only factor associated with rapid renal progression. The peripheral neutrophil count was higher in many situations prone to renal progression. We used the cutoff value of the peripheral neutrophil count 2760/mm3, divided our study population into higher and lower peripheral neutrophil count groups, and found that higher neutrophil populations had significantly higher eGFR reduction percentage than the lower group. Peripheral neutrophil count with a cutoff value of 2760/mm3 to predict high risk for rapid renal progression may be a useful and reliable tool in our clinical practice.

The CKD patients had worse clinical outcomes as compared with non-CKD patients. Hyperinflammation is considered to be one of the most important reasons. In our study, we found that hyperinflammation was noted in CKD patients. Those CKD patients had higher levels of hs-CRP and total and differential leukocyte counts. The reason for hyperinflammation in CKD is not easy to clarify whether it is a trigger or a result of a chronic underlying condition. However, we found that peripheral leukocyte counts, especially neutrophil counts, were

significantly associated with rapid renal progression. This result confirmed the importance of hyperinflammation for subsequent renal progression in CKD patients and also non-CKD population. Lower baseline eGFR was associated with rapid progression in our result and many other reports [36–41]. The exact reason for worse renal outcomes in advanced renal dysfunction patients is also not clear. But, hyperinflammation in those relatively poor renal function populations occupies a certain position in renal progression. Surprisingly, we found peripheral neutrophil was stronger than baseline eGFR and proteinuria in predicting renal outcome. The peripheral neutrophil was significantly positively associated with many well-known poor prognostic factors such as co-morbidity of DM, hypertension, smoking, lower baseline eGFR, and proteinuria. This result revealed that peripheral neutrophil was a better and more comprehensive marker to predict the renal outcome.

This study's major limitation was that the peripheral total and differential leukocyte count levels were analyzed from a single sample. However, the study population with the conditions that would interfere with peripheral leukocyte counts, such as current infection, chronic infection, inflammatory diseases, connective tissue disorders, malignancy, and receiving steroid or immunosuppressant treatment, were excluded. Thus, we believed that a single sampling for measuring peripheral total and differential leukocyte count is appropriate and reliable. Besides, the follow-up period of 2 years may not be long enough. However, the result showed that peripheral neutrophil count is significantly associated with rapid progression in the follow-up period of two years, suggesting that a follow-up period of 2 years is sufficient. In addition, the number of patients with CKD was relatively small compared to the overall cohort size. However, the prevalence of CKD in the community in our country was around 11.9% in a large-scale prospective cohort study based on 462293 adults in Taiwan [42]. Our study found the prevalence of CKD was around 15.7%, with the mean age around 57 ± 13 years old, which is older than study patients of Wen et al. with the mean age around 42 years old. It is sensible that a higher prevalence of CKD was noted in elderly adults. In addition, the drug usage of the community population is not collected. Further, the information of co-morbidity of ADPKD, autoimmune disease without immune suppressant treatment, and liver cirrhosis which may associate with progression of CKD or inflammation, was not collected.

Furthermore, some conditions that may affect the CKD progression during the follow-up period, such as those with high uric acid levels, may encounter gout attacks followed by taking NSAID, which may progress the renal function. Besides, the baseline inflammation markers such as peripheral leukocyte or neutrophil cannot fully explain these complex situations, such as some autoimmune disease patients not diagnosed yet or some special condition with an autoimmune disease but cannot tolerate immunosuppressant. However, our study tried our best to adjust possible factors associated with rapid progression, including smoking, NSAID usage, uric acid levels and found that peripheral neutrophil count was the strongest predictor of the rapid kidney progression. Our study's strengths include a sample size of more than two thousand participants from the community with longitudinal follow-up. We use the peripheral neutrophil count and found the cutoff values 2760/mm3 to predict high risk for rapid renal progression with a satisfying sensitivity and specificity. We found that those higher peripheral neutrophil count populations had significantly higher eGFR reduction percentages than the lower peripheral neutrophil count group. This result confirmed that a higher peripheral neutrophil count was independently associated with rapid renal progression in CKD and non-CKD populations.

## Conclusion

The study demonstrated that hyperinflammation with higher peripheral total and differential leukocyte counts was noted in CKD patients than in the non-CKD population with a

monotonic trend. In the definition of rapid renal outcome with reduction of eGFR percentage of 30% or greater within two years, we found peripheral differential leukocyte count was a significant factor associated with rapid renal progression besides factors of the male gender, smoking, co-morbidity of DM, hypertension, baseline eGFR levels. Moreover, the peripheral neutrophil count was the only independent factor associated with rapid renal progression after multivariable regression analysis. The optimal cutoff point of the peripheral neutrophil count of values of 2760/mm$^3$ is good for determining the high-risk population for rapid renal progression with a satisfying sensitivity and specificity.

## Acknowledgments

The authors wish to express their deepest gratitude to all the patients who participated in this study.

## Author Contributions

**Conceptualization:** Chiung-Hui Yen, Chun-Yu Chen, Heng Jung Hsu.

**Data curation:** I-Wen Wu, Chin-Chan Lee, Chiao-Yin Sun, Chun-Yu Chen, Heng-Chih Pan, Heng Jung Hsu.

**Formal analysis:** I-Wen Wu, Chin-Chan Lee, Kuang-Hung Hsu, Heng-Chih Pan, Heng Jung Hsu.

**Investigation:** Chin-Chan Lee, Kuang-Hung Hsu, Chiao-Yin Sun.

**Methodology:** Kuang-Hung Hsu, Chun-Yu Chen.

**Resources:** Chiao-Yin Sun, Chun-Yu Chen.

**Supervision:** Heng Jung Hsu.

**Validation:** Heng Jung Hsu.

**Writing – original draft:** Chiung-Hui Yen.

**Writing – review & editing:** Heng Jung Hsu.

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
