## [Decision Letter · Decision Letter 0]

4 Jul 2021

PONE-D-21-09532

The Prognostic Value of Peripheral Total and Differential Leukocyte Count in Renal Progression: A Community-Based Study

PLOS ONE

Dear Dr. Heng Jung Hsu,

Thank you for submitting your manuscript to PLOS ONE. After careful consideration, we feel that it has merit but does not fully meet PLOS ONE’s publication criteria as it currently stands. Therefore, we invite you to submit a revised version of the manuscript that addresses the points raised during the review process.

We look forward to receiving your revised manuscript.

Kind regards,

Ping-Hsun Wu, M.D. PhD.

Academic Editor

PLOS ONE

Additional Editor Comments:

Short observational time may bias the study results. Besides, CKD patients with subgroup analysis were suggested to understand the association between total and differential leukocyte count and renal function progression in different etiology of CKD. In particular, some glomerulonephritis or autoimmune disease-related CKD may influence the baseline differential leukocyte count.

Journal Requirements:

[This investigation was partially supported by a grant from Chang Gung Medical Foundation Chang Gung Memorial Hospital, Keelung (CMRPG260362-3, CMRPG2A0451-3,CLRPG2J0011,CMRPG2B0151-3,and CMRPG2E0251-3).]

 [The funders had no role in study design, data collection and analysis, decision to publish, or preparation of the manuscript.]

Reviewers' comments:

Reviewer's Responses to Questions

**Comments to the Author**

1. Is the manuscript technically sound, and do the data support the conclusions?

Reviewer #1: Yes

Reviewer #2: No

2. Has the statistical analysis been performed appropriately and rigorously? 

Reviewer #1: Yes

Reviewer #2: N/A

3. Have the authors made all data underlying the findings in their manuscript fully available?

Reviewer #1: No

Reviewer #2: No

4. Is the manuscript presented in an intelligible fashion and written in standard English?

Reviewer #1: No

Reviewer #2: No

5. Review Comments to the Author

Reviewer #1: The work presented is of interest in that it demonstrates an association between neutrophil count in CKD and the progression of CKD in a large cohort of patients. The enrolment and follow-up of >2000 patients is a strength of the study. Many of the results presented in the earlier sections of the paper e.g. increased phosphate, increased uric acid, lower haemoglobin in CKD are expected based on existing clinical knowledge.

The limitations of the work are largely acknowledged by the authors i.e. the relatively short duration of follow-up and the fact that clinical information such as drug use and co-morbidities may be incomplete and may have confounded results. It should also be noted that the number of patients with CKD was relatively small, compared to the overall cohort size.

I do feel that the grammar should be improved throughout the manuscript prior to publication.

I have a number of other minor comments:

- In the introduction the acronym MIA is used but not explained - suggest clarifying

- Suggest giving more detail about exclusion criteria especially as regards co-morbidities which led to exclusion

- Suggest clarifying method of measuring proteinuria

- Consider making the discussion section more concise and citing some additional works regarding the association between neutrophil count and prognosis in renal disease if possible. The work of Bowe et al. ( https://doi.org/10.2215/CJN.09710916) would also be relevant

-It is unclear to me whether the dataset has been made public and whether it would be required by PLOS ONE in this instance

Reviewer #2: The analysis for the patient inclusion is not suitable while their baseline disease may also with lots of comorbidity which interference WBC count. Only exclude patient with malignancy, steroid use or immune suppressant is not enough.

Please analysis patient into sub-group (ex: GN, ADPKD, autoimmune disease without immune suppressant treatment, Liver cirrhosis...).

High uric acid level patient may also with gout and these patient may take NSAID which progress the renal function then how to point the leukocyte differential ratio explanation to above condition in the inclusion case.

Some autoimmune disease patient may hide in the inclusion patient which no doctor noted them or some lupus patient cannot use steroid or immune suppressant because contraindication to their comorbidity which would have WBC difference compared to other CKD patient, then the result from the study with high limitation to explain this part.

6. PLOS authors have the option to publish the peer review history of their article (what does this mean?). If published, this will include your full peer review and any attached files.

Reviewer #1: No

Reviewer #2: No

---

## [Author Response · Author response to Decision Letter 0]

17 Jul 2021

Title: The prognostic value of peripheral total and differential leukocyte count in renal progression: A community-based study

Authors: Chiung-Hui Yen, I-Wen Wu, Chin-Chan Lee, Kuang-Hung Hsu, Chiao-Yin Sun, Chun-Yu Chen, Heng-Chih Pan, Heng Jung Hsu

Manuscript No. PONE-D-21-09532

Answers to Academic Editor

1. The Editor indicated that funding information should not appear in the “Acknowledgments” section or other areas of the manuscript. Thanks for Editor's reminder and we had removed the statements about funding information within the manuscript and added the fund information within the cover letter. Would you please help me to update the "Funding Statement” section of the online submission form? 

The Funding statements are as following:

This investigation was partially supported by a grant from Chang Gung Medical Foundation Chang Gung Memorial Hospital, Keelung (CMRPG260362-3, CMRPG2A0451-3, CLRPG2J0011, CMRPG2B0151-3, and CMRPG2E0251-3). The funders had no role in study design, data collection and analysis, decision to publish, or preparation of the manuscript. All the funding of support was received during this study. There was no additional external funding received for this study.

2. The Editor indicated the ethics statement should only appear in the “Methods section” of the manuscript. We agree that the ethics statement appears in the “Methods section” of the manuscript and removed the ethics statements written in any section besides the “Methods” section. 

3. The Editor suggested subgrouping the CKD patients into different CKD etiology to analyze the association between total and differential leukocyte count and renal function progression. We agree that CKD patients with different CKD etiology, such as glomerulonephritis or autoimmune disease-related CKD may influence the baseline differential leukocyte count. We had re-analysis our CKD patients and subgrouping them by different CKD etiology and found the levels of hs-CRP were similar between different primary causes of CKD, but significantly higher levels of peripheral leukocyte count, neutrophil count, and eosinophil count were noted in diabetic nephropathy than other primary causes of CKD. However, the percentage of rapid renal progression was similar between different primary causes of CKD (p = 0.569). Besides, rapid renal progression was positively and independently associated with the peripheral neutrophil count after adjusting male gender, co-morbidity of DM, and uric acid levels. The above statements about subgrouping CKD patients with different etiology were added in the “Result” part of the revised manuscript. (Page 13, line 12-16 and Page 14, line 11-13 )

Lots of thanks to the Editor for reviewing our manuscript. We think our manuscript will be better after your expert suggestion.

 

Title: The prognostic value of peripheral total and differential leukocyte count in renal progression: A community-based study

Authors: Chiung-Hui Yen, I-Wen Wu, Chin-Chan Lee, Kuang-Hung Hsu, Chiao-Yin Sun, Chun-Yu Chen, Heng-Chih Pan, Heng Jung Hsu

Manuscript No. PONE-D-21-09532

Answers to Reviewer 1

1. The reviewer indicated that the number of study patients with CKD was relatively small compared to the overall cohort size. We agree that the number of patients with CKD was relatively small compared to the overall cohort size. However, the prevalence of CKD in the community in our country was around 11.9% in a large-scale prospective cohort study based on 462293 adults in Taiwan [42]. Our study found the prevalence of CKD was around 15.7%, with the mean age around 57 ± 13 years old, which is older than study patients of Wen et al. with the mean age around 42 years old. It is sensible that a higher prevalence of CKD was noted in elderly adults. The above statements discussing relatively small CKD patients' numbers were added in the "Discussion" part of the revised manuscript. (Page 22, line 15- 22)

[42]. Wen CP, Cheng TY, Tsai MK, Chang YC, Chan HT, Tsai SP, et al. All-cause mortality attributable to chronic kidney disease: a prospective cohort study based on 462 293 adults in Taiwan. Lancet. 2008;371(9631):2173-82. Epub 2008/07/01. doi: 10.1016/S0140-6736(08)60952-6. PubMed PMID: 18586172.

2. The reviewer indicated the grammar should be improved throughout the manuscript. We agree that the grammar in our manuscript is not good enough and had corrected grammar errors throughout the manuscript. In addition, we also request one friend with the native language of English to check the revised manuscript. We think the grammar in the revised manuscript may be suitable for publication. 

3. The reviewer suggested that the acronym MIA in the “Introduction” part should be explained and clarified. We agree that the acronym MIA should be clarified to explain the complex relationship between malnutrition, inflammation, and atherosclerosis in CKD. CKD patients with dialysis had a higher prevalence of protein-energy malnutrition and inflammation, with pathophysiology related to nutrient loss, increased protein catabolism, and hypoalbuminemia. Proinflammatory cytokines like interleukin -1 and tumor necrosis factor-alpha play a major role in the onset of metabolic alterations in CKD patients. Whether inflammation is either a trigger or a result of CKD is still complicated. However, many elements contribute to the inflammatory status of CKD, including increased production of proinflammatory cytokines, oxidative stress and acidosis, chronic and recurrent infections, altered metabolism of adipose tissue, and gut microbiota dysbiosis. The above statements to explain the acronym MIA are added in the “Introduction” part of the revised manuscript. (Page 5, line 7-16)

4. The reviewer indicated that the exclusion criteria in the study should be more detailed, especially regarding co-morbidities. We agree that the inclusion and exclusion criteria of the study are essential for the study. The exclusion criteria were conditions that would interfere with peripheral leukocyte counts, such as current infection-related admission, chronic infection due to tuberculosis, arthritis, inflammatory bowel diseases, connective tissue disorders, other inflammatory diseases, or malignancy. The statements about exclusion criteria in the study were added in the “Materials and Methods” part of the revised manuscript. (Page 7, line 6-10)

5. The reviewer indicated the method of measuring proteinuria should be clarified. We agree the method of measuring proteinuria should be clarified in the study. Daily proteinuria amount was estimated by spot urine protein-to-creatinine ratio (mg/g) according to the NKF/DOQI [22]. We had added the above statement in the “Materials and Methods” part of the revised manuscript. (Page 8, line 1-2)

[22]. Levey AS, de Jong PE, Coresh J, El Nahas M, Astor BC, Matsushita K, et al. The definition, classification, and prognosis of chronic kidney disease: a KDIGO Controversies Conference report. Kidney Int. 2011;80(1):17-28. Epub 2010/12/15. doi: 10.1038/ki.2010.483. PubMed PMID: 21150873.

6. The reviewer suggested making the discussion section more concise and citing some additional works regarding the association between neutrophil count and prognosis in renal disease. We agree that making a more concise discussion and citing more additional works regarding the association between neutrophil count and prognosis in renal disease in the "Discussion" part may make the study more comprehensive. Bowe et al. also evaluated the association between peripheral leukocyte count and risk of incident CKD and progression to ESRD [34]. This study enrolled more than 1594700 United States veterans with a median follow-up duration of 9.2 years and found that those with higher monocyte count had significantly higher risks of incident CKD and CKD progression to ESRD. Although our study did not enroll as many populations as Bowe et al. and the follow-up duration of two years was relatively not long enough, our study also found the peripheral monocyte counts were associated with rapid progression of the kidney. However, the association loss significance after multivariate analysis and only peripheral neutrophil counts was independently significantly associated with rapid progression of kidney. Due to Bowe et al. did not evaluate the role of peripheral neutrophils in the association of renal progression, the result of positive connection between monocyte counts and incident CKD and progression to ESRD is reasonable. However, the result of our study and Bowe et al. all suggested that inflammatory cell infiltrates may play a role in the development and progression of kidney disease. The above statements were added in the “Discussion” part of the revised manuscript. (Page 19, line 5-19)

[34]. Bowe B, Xie Y, Xian H, Li T, Al-Aly Z. Association between Monocyte Count and Risk of Incident CKD and Progression to ESRD. Clin J Am Soc Nephrol. 2017;12(4):603-13. Epub 2017/03/30. doi: 10.2215/CJN.09710916. PubMed PMID: 28348030; PubMed Central PMCID: PMCPMC5383390.

7. The reviewer indicated if the dataset of the study has been made public. In our study, all data generated or analyzed during this study are included in this published article and have been made public. The above statements are noted in the revised manuscript. (Page 2, line 7-8)

Lots of thanks to the reviewer for reviewing our manuscript. We think our manuscript will be better after your expert suggestion.

 

Title: The prognostic value of peripheral total and differential leukocyte count in renal progression: A community-based study

Authors: Chiung-Hui Yen, I-Wen Wu, Chin-Chan Lee, Kuang-Hung Hsu, Chiao-Yin Sun, Chun-Yu Chen, Heng-Chih Pan, Heng Jung Hsu

Manuscript No. PONE-D-21-09532

Answers to Reviewer 2

1. The reviewer indicated the included patients with lots of co-morbidity, which interference WBC count should be clarified. We agree that patients with lots of co-morbidity which interference WBC count should be clarified. The exclusion criteria were conditions that would interfere with peripheral leukocyte counts, such as current infection-related admission, chronic infection due to tuberculosis, arthritis, inflammatory bowel diseases, connective tissue disorders, other inflammatory diseases, or malignancy. Besides, CKD patients with different CKD etiology, such as glomerulonephritis or autoimmune disease-related CKD, may influence the baseline differential leukocyte count. We had re-analysis our CKD patients and subgrouping them by different CKD etiology and found the levels of the hs-CRP were similar between different primary causes of CKD, but significantly higher levels of peripheral leukocyte count, neutrophil count, and eosinophil count were noted in diabetic nephropathy than other primary causes of CKD. However, the percentage of rapid renal progression was similar between different primary causes of CKD (p = 0.569). Besides, rapid renal progression was positively and independently associated with the peripheral neutrophil count after adjusting male gender, co-morbidity of DM, and uric acid levels. The above statements about exclusion and subgrouping CKD patients with different etiology were added in the “Materials and Methods” part and “Result” part of the revised manuscript. (Page 7, line 6-10) (Page 13, line 12-16) (Page 14, line 11-13)

2. The reviewer suggested analyzing patients into sub-group (ex: GN, ADPKD, autoimmune disease without immune suppressant treatment, Liver cirrhosis) to evaluate the association between total and differential leukocyte count and renal function progression. We agree to sub-grouping study patients with different CKD etiology in Table 3 of the revised manuscript. However, the co-morbidity of ADPKD, autoimmune disease without immune suppressant treatment, and liver cirrhosis which may associate with progression of CKD or inflammation, was not collected. The statements about lacking the information about co-morbidity of ADPKD, autoimmune disease without immune suppressant treatment, and liver cirrhosis were added in the “Discussion" part of the revised manuscript. (Page 23, line 1-3)

3. The reviewer indicated some situations such as patients with high uric acid levels or autoimmune disease patients without diagnosis or some lupus patients could not use steroids or immunosuppressants because contraindication would have WBC difference compared to other CKD patients. We agree that the study cannot exclude some autoimmune disease patients who weren't diagnosed yet or some special condition with an autoimmune disease but cannot tolerate immunosuppressants. some conditions that may affect the CKD progression during the follow-up period, such as those with high uric acid levels, may encounter gout attacks followed by taking NSAID, which may progress the renal function. Besides, the baseline inflammation markers such as peripheral leukocyte or neutrophil cannot fully explain these complex situations, such as some autoimmune disease patients not diagnosed yet or some special condition with an autoimmune disease but cannot tolerate immunosuppressant. However, our study tried our best to adjust possible factors associated with rapid progression, including smoking, NSAID usage, uric acid levels and found that peripheral neutrophil count was the strongest predictor of the rapid kidney progression. The above statements about the limitation of our study were added in the "Discussion" part of the revised manuscript. (Page 23, line 4-13)

Lots of thanks to the reviewer for reviewing our manuscript. We think our manuscript will be better after your expert suggestion.

---

## [Decision Letter · Decision Letter 1]

22 Sep 2021

The Prognostic Value of Peripheral Total and Differential Leukocyte Count in Renal Progression: A Community-Based Study

PONE-D-21-09532R1

Dear Dr. Heng Jung Hsu,

We’re pleased to inform you that your manuscript has been judged scientifically suitable for publication and will be formally accepted for publication once it meets all outstanding technical requirements.

Kind regards,

Ping-Hsun Wu, M.D. PhD.

Academic Editor

PLOS ONE

Additional Editor Comments (optional):

All comments had been responded accordingly. No further comments were suggested by reviewer.

Reviewers' comments:

Reviewer's Responses to Questions

**Comments to the Author**

1. If the authors have adequately addressed your comments raised in a previous round of review and you feel that this manuscript is now acceptable for publication, you may indicate that here to bypass the “Comments to the Author” section, enter your conflict of interest statement in the “Confidential to Editor” section, and submit your "Accept" recommendation.

Reviewer #3: All comments have been addressed

2. Is the manuscript technically sound, and do the data support the conclusions?

Reviewer #3: Yes

3. Has the statistical analysis been performed appropriately and rigorously? 

Reviewer #3: Yes

4. Have the authors made all data underlying the findings in their manuscript fully available?

Reviewer #3: Yes

5. Is the manuscript presented in an intelligible fashion and written in standard English?

Reviewer #3: Yes

6. Review Comments to the Author

Reviewer #3: Using community-based survey data, Yen et al. investigated the association between peripheral total and differential leukocyte count and renal progression. The author replied to all the comments from reviewers. I have no further suggestions for this paper.

7. PLOS authors have the option to publish the peer review history of their article (what does this mean?). If published, this will include your full peer review and any attached files.

Reviewer #3: No

---

## [Editor Report · Acceptance letter]

20 Oct 2021

PONE-D-21-09532R1 

The Prognostic Value of Peripheral Total and Differential Leukocyte Count in Renal Progression: A Community-Based Study 

Dear Dr. Hsu:

I'm pleased to inform you that your manuscript has been deemed suitable for publication in PLOS ONE. Congratulations! Your manuscript is now with our production department. 

Kind regards, 

on behalf of

Dr. Ping-Hsun Wu 

Academic Editor

PLOS ONE